# System biology and network-based approach to identify the therapeutic signatures and potential inhibitors against polycystic lipomembranous osteodysplasia with Sclerosing Leukoencephalopathy

Bayan T. Bokhari[1]*, Alaa M. Saleh[1], Hashim M. Aljohani[2,3,4], Wejdan Hussain Owaydhah[5], Mohammad Ahmad Alobaidy [6], Naief Dahran [7], Hind M. Naffadi[8], Alaa Abdulaziz Eisa [2,4]*

1 Department of Clinical Laboratory Sciences, Faculty of Applied Medical Sciences, Umm Al-Qura University, Makkah, Saudi Arabia, 2 Department of Clinical Laboratory Sciences, College of Applied Medical Sciences, Taibah University, Medina, Saudi Arabia, 3 Department of Pathology and Laboratory Medicine, College of Medicine, University of Cincinnati, Cincinnati, Ohio, United States of America, 4 Health and Life Research Center, Taibah University, Medina, Saudi Arabia, 5 Department of Basic Medical Sciences, Faculty of Medicine, Taibah University, Madinah, Saudi Arabia, 6 Department of Anatomy, Faculty of Medicine, Umm Al-Qura University, Makkah, Saudi Arabia, 7 Department of Basic Medical Sciences, College of Medicine, University of Jeddah, Jeddah, Saudi Arabia, 8 Department of Medical Genetics, Faculty of Medicine, Umm Al-Qura University, Makkah, Saudi Arabia

* btbokhari@uqu.edu.sa (BTB); aeisa@taibahu.edu.sa (AAE)

## Abstract

Polycystic Lipomembranous Osteodysplasia with Sclerosing Leukoencephalopathy (PLOSL), often recognized as Nasu-Hakola disease, is an uncommon autosomal recessive systemic condition described by numerous bone lesions that resemble cysts and progressive early-onset dementia. One novel way to find possible drug targets is to use Network Analyst tools for network-based gene expression profiling. Identifying the target hub genes, essential for the initiation and progression of PLOSL, is validated by the significance level (p-score) attained using Cytoscape in the survival analysis of the major central genes. The X2K online tool also examined the regulatory kinases that formed the interaction between protein molecule networks. Out of the 53 genes obtained to be differentially expressed, PPARG and AIF1 had the greatest degree score, followed by C1QA with 5 degrees and SIGLEC1 and MSR1 with 4 degrees. Furthermore, Molecular docking of target PPARG gene with AMG-131 and Elafibranor drugs, having the chemical formulas 2-[2,6-dimethyl-4-[(E)-3-(4-methylsulfanylphenyl)-3-oxoprop-1-enyl]phenoxy], and 3,5-dichloro-4-quinolin-3-yloxyphenyl) benzenesulfonamide, along control (Rosiglitazone (S) shows binding affinities of −7.2 kcal/mol, −7.4 kcal/mol, and −7.6 kcal/mol respectively. The enzyme remained extremely stable in the complex throughout 200 ns, with a mean Root Mean Square Deviation (RMSD) of 2.95 Å for the AMG-131 complex system and 2.93 Å against the Elafibranor complex system. Root Mean Square

**Data availability statement:** All relevant data are within the manuscript and its Supporting Information files.

**Funding:** The author(s) received no specific funding for this work.

**Competing interests:** The authors have declared that no competing interests exist.

Fluctuation (RMSF) anticipated steady behavior with average RMSD for the active site residues in the docked system. Arg76 and Leu28leu Leu118 were shown to be essential enzyme residues for binding, anchoring, and bridging strong hydrogen and hydrophobic interactions between the enzyme and the inhibitor, according to the Radial Distribution Function (RDF). These results broadened our knowledge of putative biomarkers for PLOSL diseases, and an experimental strategy will improve our results even more in the future.

## 1. Introduction

Nasu-Hakola disease, also known as polycystic lipomembranous osteodysplasia with sclerosing leukoencephalopathy (PLOSL), is an uncommon autosomal recessive condition marked by a combination of irreversible dementia with early onset and polycystic bone degradation [1]. Due to its strong genetic and hereditary base, the disease is more common in some populations, such as Finnish and Japanese groups [2]. Proper phagocytosis and osteoclast proliferation are disrupted by mutations in TREM2 or TYROBP, which are crucial regulators of immunological signaling and microglial function. This results in severe neurological and muscular signs characteristic with PLOSL, as well as neuroinflammation and skeletal fragility [1,3,4]. It is crucial to comprehend the disease's molecular underpinnings and find therapeutic targets because it progresses quickly and the only treatments available are supportive and symptomatic care [5].

Drug research is still expensive and time-consuming; typical pipelines need a significant financial investment and more than ten years [6,7]. While early-stage discovery capacity has increased because to advancements in combinatorial chemistry and high-throughput screening methods, authorized therapies have not increased proportionately [8]. This emphasizes the necessity for alternate tactics that can rank potential targets in order of importance and weed out unsuitable applicants early on [9]. Computational methods, such as structure-based drug design (SBDD) and computer-aided drug design (CADD), provide effective tools for assessing molecular interactions, forecasting pharmacological response, and directing early discovery efforts [10,11]. Simultaneously, systems biology integrates transcriptome, proteomic, and network-level data to provide a strong foundation for comprehending complicated disorders.

In this work, we characterize the molecular landscape of PLOSL using a combined systems biology and computational modeling approach. We want to uncover important genes, pathways, and possible medicinal drugs using molecular docking, molecular dynamics simulations, protein–protein interaction (PPI) network modeling, and differential expression analysis. By generating novel hypotheses and emphasizing prospects for future experimental validation, this integrated strategy aims to address the existing paucity of tailored treatments for this uncommon and crippling condition.

## 2. Materials and methods

This study incorporated an integrative approach combining systems biology and network analysis, as outlined in **Fig 1**, to elucidate molecular pathways and potential therapeutic targets for PLOSL.

### 2.2. Data acquisition for comparative analysis

We sourced our GSE3624 dataset from the NCBI's GEO database (https://www.ncbi.nlm.nih.gov/geo/), a repository for high-throughput genomic data [12]. This dataset comprised eight samples, three from normal individuals and five from PLOSL patients, all analyzed on the [HG-U133A] Affymetrix Human Genome U133A Array platform [13].

### 2.3. Delineation and screening of Differentially Expressed Genes (DEGs)

The Network Analyst platform (https://www.networkanalyst.ca/) evaluated gene expression variations [14]. Comparing the expression patterns of bone marrow cells from patients with PLOSL with normal bone marrow cells proved to be a helpful application of this technology. The selection of genes was based on statistically significant differences. The data matrix was set up so that separate rows represented different genes and corresponding columns represented samples. Three control samples and five PLOSL patient samples were included in this matrix. To ensure consistency and facilitate analysis, gene identifiers were methodically transformed into Entrez IDs [15].

A Network Analyst for a meta-analysis of microarray data was used, applying Variance Stabilizing Normalization (VSN) using maximum likelihood estimates and parametric modifications to align sample variances regardless of their mean intensities. Log2 and quantile normalization were also used, with box plots and PCA plots verifying their effectiveness [16]. Differentially expressed genes (DEGs) were identified using strict criteria: adjusted p-value (FDR) < 0.05 (Benjamin-Hochberg method), a t-test based on Pearson correlation and the Limma algorithm to investigate the relationships

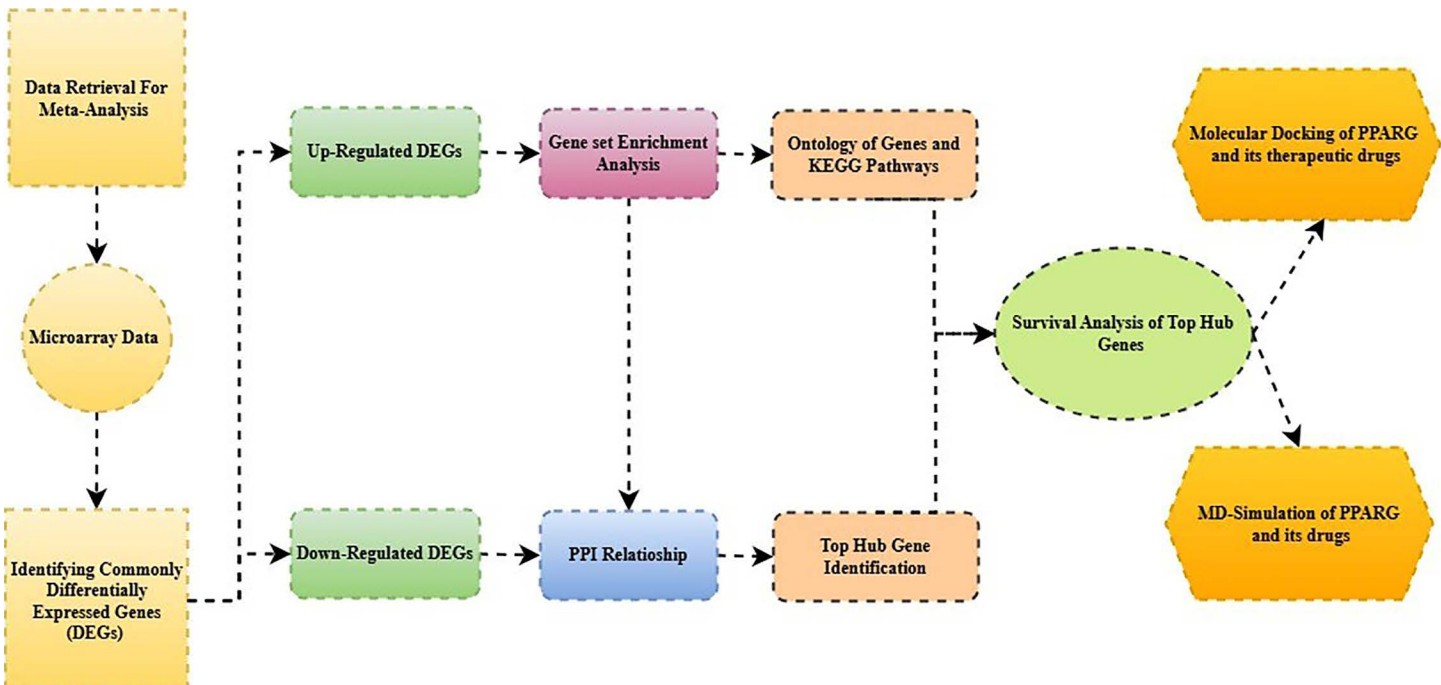

**Fig 1. Systematic representation of the integrated bioinformatics analysis utilized in this study.**

between various variables [17]. Compared to normal controls, this multimodal approach yielded a comprehensive list of DEGs in PLOSL samples.

## 2.4. Integration of gene ontology and pathway enrichment

For the gene set functional enrichment study, Gene ontology (GO) components and KEGG pathway linkages were the main areas of interest which were conducted using TNMplot and Network Analyst [14]. These investigations made the molecular and cellular processes impacted by PLOSL clearer. We have used distinct subgroups of GSE3624, which had eight samples in total: Three from normal individuals and five from PLOSL patients. To find differentially expressed genes (DEGs) that were pertinent to our goal, the full dataset was processed using the Network Analyst tool. To evaluate the KEGG pathway and Gene Ontology (GO) enrichment of differentially expressed genes (DEGs) with a significant level of $p < 0.05$, the free bioinformatics tools TNMplot (https://tnmplot.com/) and NetworkAnalyst (https://www.networkanalyst.ca/) were utilized [18]. DEG modules and central genes were uploaded for functional and pathway enrichment analysis, considering a number of parameters, including cell components, molecular functions, biological processes, function mode, gene ID, species selection (Homo sapiens), and supporting information from pertinent GO and KEGG databases. Large-scale genomic or transcriptional datasets can be interpreted for their functional roles using the well-known Gene Ontology (GO) analysis approach [19].

## 2.5. PPI network generation

The STRING database (https://string-db.org/) [20] was targeted to map the interactions of proteins linked to the identified DEGs in PLOSL, which contains data on over 67 million proteins across approximately 14,000 species. Cytoscape 3.9.1 visualized these molecular networks and biological pathways [21], allowing us to import the interaction data from STRING. A confidence threshold of 0.75 was set, ensuring medium confidence in the network's validity [18,19], focusing on the original DEGs. The entire dataset, encompassing these genes, was integrated into Cytoscape [22]. To avoid complexity, zero-order interaction was used to prevent overwhelming complexity, commonly known as the 'hairball' effect [23]. The "Network Analyzer" feature in Cytoscape further assessed parameters like network density, node degree distribution, clustering coefficient, power law distribution of nodes, and network centralization [24], providing insights into key protein interactions relevant to PLOSL's pathophysiology.

## 2.6. Key hub proteins identification in the PPI network

The Cytoscape (https://cytoscape.org/), a sophisticated tool, was used to visualize molecular interaction networks and identify key hub proteins in the PPI network [22]. Protein centrality was calculated based on the number of connections or edges that a protein shared with other proteins in the built PPI network. These hub genes are important and potentially crucial to PLOSL's molecular mechanisms. Hub gene candidates were identified as proteins with a high degree of connectedness, which indicated a significant number of interactions within the network [25]. Incorporating eleven distinct algorithms, the CytoHubba plugin in Cytoscape helped identify critical nodes and narrow the focus to the most significant genes in our PPI network [26]. Due to their significant connectivity, these central hub proteins are hypothesized to be vital in the pathophysiology of PLOSL and, therefore, might emerge as promising targets for therapeutic interventions.

## 2.7. Analysis of regulatory associations and key transcription factors

Key transcription factors (TFs) and kinases were identified using the Expression2Kinases tool (https://maayanlab.cloud/X2K/), influencing the identified DEGs. The entire array of DEGs, complete with specified gene markers, was uploaded to the X2K system for analysis [27]. This tool adeptly calculated the ten most prominent transcription factor regulators and associated kinases, utilizing its kinase module and transcription factor (TF) analysis. The assessment was grounded on the Fischer test p-values, ensuring a robust statistical framework for the findings. X2K facilitated the construction of a

comprehensive regulatory network by integrating data from the ChEA69 database, which is rich in ChIP-seq-derived information. The resulting network was shown using Cytoscape's capability to read and display "graphml" files. This dynamic model visualization captured the intricate interactions between different transcription factors and kinases, not just a static depiction [27].

## 2.8. Survival analysis of hub genes

The Kaplan-Meier plotter program (https://kmplot.com/), a versatile online tool [28] generates survival curves for each hub gene, demonstrating the relationship between the expression of genes and survival rates of patients. This approach provides a mathematical and visual representation of survival probabilities over time, aiding in identifying potential prognostic biomarkers for PLOSL.

## 2.9. Protein-drug interaction network

The possible medication interactions with top hub genes using Drug Bank (version 5.1.10) was used, an online database integrated with Network Analyzer. This analysis included FDA-approved drugs and investigational compounds. This extensive database encompasses many substances ranging from FDA-approved drugs to compounds that are still in the investigational phase. The Drug Bank database is remarkably extensive, featuring a total of 15,466 drug entries. This includes 2,741 small molecule drugs that have received FDA approval, 1,577 biologics such as proteins, peptides, vaccines, and allergens, alongside 134 nutraceuticals, and a significant number of drugs (over 6,717) currently in the experimental stage. Furthermore, the database links these drug entries to 5,293 distinct non-redundant protein sequences, which serve various roles, such as drug targets, enzymes, transporters, or carriers [29].

## 2.10. Data acquisition for comparative analysis

The NCBI Gene Expression Omnibus (GEO) database was sourced from a publicly accessible gene expression data repository (https://www.ncbi.nlm.nih.gov/geo/), The datasets used in this work, GSE3624 and GSE25496. Eight samples in all three from normal individuals and five from PLOSL patients are included in the GSE3624 collection. while the GSE25496 dataset, on the other hand, includes two samples, one from a PLOSL patient and one from a healthy control. The Affymetrix Human Genome U133A Array (HG-U133A), a microarray platform frequently used for gene expression profiling, was utilized to create both datasets. Both datasets were selected as the most pertinent and easily accessible sources for this uncommon illness since they are the only publicly accessible transcriptome datasets with PLOSL patient samples and matched controls.

## 2.11. Molecular docking of PPARG gene with target drugs

To find the most effective inhibitors, molecular docking analysis was performed using PyRx software [30]. We have retrieved the target receptor PPARG involved in Lipomembranous Osteodysplasia with Sclerosing Leukoencephalopathy (PLOSL), from the protein database with PDB id: 1FM6. Receptor active sites were predicted using literature support [31]. An Arg76 hotspot residue that is thought to be critically important for catalytic activity was used as the coordinates for docking against the top 2 compounds, i.e., AMG-131 and Elafibranor. The dimensions of the binding site at 15 Å, which had an X-axis of 16.777, Y-axis of −0.086, and Z-axis of 3.233, were determined by pointing coordinates of the atom Arg76. This has been followed by positive and negative control inhibitors with other FDA approved drugs, i.e., Positive Control- Rosiglitazone (S), Samarium Sm-153 Lexidronam, Clemastine, and Negative control Acetaminophen against their respected targets. Eight iterations were allocated to each compound, and the compound that achieved the largest negative binding energy (kcal/mol) and the most stable binding conformer was ranked first. MD simulations were performed for both complex systems [33].

## 2.12. Molecular Dynamics (MD) simulation

The molecular docking prediction regarding the lead compounds achieved during the network pharmacological study with its binding mode was assessed using MD simulation. The trajectories generated during the MD simulation production phase, the stability data and the profile of chemical interactions of the MD system were recovered. MD simulations were run using the AMBER 16 simulation program [32]. The Antechamber software produced the MD system parameters library [33]. The compounds' parameters were created using the General Amber Force Field (GAFF) [34] whereas, FF14SB force field was used for protein [35], while the AMBER leap module [36] helped center the complex in a 12×TIP3P water box [37]. The system's charge was neutralized by adding the proper counter ions. The energy of the system was gradually reduced through multiple rounds: 100 steps were dedicated to minimizing hydrogen atoms, 500 steps to reducing water boxes, 500 steps to minimizing the entire system while imposing a 5 kcal/mol Å$^2$ constraint on Cα, and 100 steps were devoted to minimizing non-heavy atoms while maintaining a 10 kcal/mol Å$^2$ constraint. Next, an NVT ensemble was used to gradually heat the system from 0 to 300 K over 20 ps, with a time step of 2 femtoseconds and a constraint of 5 kcal/mol –Å$^2$ on systems Cα. While the SHAKE algorithm [38], a constant hydrogen bond length and langevin dynamics [40] were maintained to accomplish temperature hold. Then, using a time step of two femtoseconds, the systems were adjusted for 1 ns. After that, each system's constant temperature and pressure (NPT) ensemble [39] was created by running the system for 1 ns at 300 K and 1 bar of pressure. At last, a 1 ns equilibration period was granted to the systems. For the lead compound, a 200-ns production run was carried out. Using a time step of 2 fs and non-bound interactions set to 8 Å, herein the production phase was initiated within NVT ensemble, where a Berendsen temperature coupling method was used to maintain a thermal stability constant temperature [40]. The CPPTRA module was utilized to conduct simulation trajectories to investigate structural systems [41].

## 3. Results

### 3.1. Quality control and principal component analysis

Expression profiling was performed for another data set for comparative analysis using the PLOSL gene datasets GSE3624 and GSE25496 from GEO (NCBI), with each subject detail presented in **Table 1** and **Table 2**. To identify DEGs, p-values and log2 fold change (log2FC) values were used by the Network Analyst tool, considering genes with logFC > 1 or < −1 and p-value < 0.05 as significant, with DEGs calculated using the predetermined cutoff criteria. Quantile normalization like variance-stabilizing normalization, was applied to assess transcriptome expression, and normalization results are shown in S1 Fig. Principal component analysis (PCA) identified unique groupings based on gene transcription levels in mutant and control samples.

### 3.2. Differential expression analysis and PPI network analysis

A thorough study was conducted on the normal and unhealthy groups of both the data set GSE3624 and GSE25496 using several statistical tests, such as the student's t-test, the Pearson correlation test and the Benjamin-Hochberg technique. The investigations conducted yielded a list of 53 DEGs in GSE3624 data set while 20 DEGs in GSE25496. Among these discovered DEGs, 41 genes showed overexpression, and 12 genes showed downregulation in GSE3624 data set, on other hand 8 genes show upregulation and 12 genes show downregulation in GSE25496 data set were finally prioritized

**Table 1. Displays the data related to the samples in the GSE3624 dataset that were retrieved from the Gene Expression Omnibus database.**

| Group | Accession | Organism | Disease State | Cell type |
|---|---|---|---|---|
| Normal | GSM626950 | Homo sapiens | Normal | The control brain RNA |
| PLOSL | GSM626951 | Homo sapiens | PLOSL | The Nasu-Hakola disease brain RNA |

**Table 2. Presents data pertaining to the samples in the GSE25496 dataset obtained from the Gene Expression Omnibus database.**

| Group | Accession | Organism | Disease State | Cell type |
|---|---|---|---|---|
| Normal | GSM83892 | *Homo sapiens* | Normal | Monocyte-derived DCs |
| | GSM83894 | *Homo sapiens* | Normal | Monocyte-derived DCs |
| | GSM83894 | *Homo sapiens* | Normal | Monocyte-derived DCs |
| PLOSL | GSM83889 | *Homo sapiens* | PLOSL | Monocyte-derived DCs |
| | GSM83890 | *Homo sapiens* | PLOSL | Monocyte-derived DCs |
| | GSM83891 | *Homo sapiens* | PLOSL | Monocyte-derived DCs |
| | GSM83895 | *Homo sapiens* | PLOSL | Monocyte-derived DCs |
| | GSM83896 | *Homo sapiens* | PLOSL | Monocyte-derived DCs |

based on the changed or corrected p-value. A key strategy for revealing the complexities of cellular networking systems is to investigate the PPI network. The degree of the ailment can be inferred from differences seen in the protein networks of cells between the normal and unhealthy groups. Nodes in this computer model represent genes and proteins, while edges show the links or connections between these genes and proteins, by mapping the discovered DEGs, a network representing the relationships between them was created. Cytoscape was used to display the visualization of these discovered DEGs. A moderate confidence level of 0.75 was used for both the data sets to query the String repository to obtain interactions between DEGs. For GSE3624 dataset there were 28 nodes and 27 edges in the network. Calculations revealed that the mean local clustering coefficient was 0.377, and the mean node degree was 1.02. On the other hand, For GSE25496 dataset there were 12 nodes and 20 edges in the network, local clustering coefficient was 0.186 and the mean node degree was 1. Furthermore, Genes were represented as nodes in this study, with edges representing their connections. The differentially expressed genes (DEGs) discovered in the investigation were used to design an interaction network. These DEGs were mapped, and their associations were displayed with Cytoscape. Data for these associations were acquired from the STRING database, with a moderate confidence level of 0.75. These measures show the significance of the relationships inside our network in **Fig 2** and S2 in S1 Fig. In addition, **Fig 3** and S3 in S1 Fig depicts the DEGs visually using a volcano plot and a heat map. The volcano plot demonstrates the statistically significant and fold-change pattern of the DEGs, whilst the heatmap depicts expression patterns across samples, allowing for clear separation between upregulated and downregulated genes. These results, taken together, provide evidence for the dependability and biological importance of the detected DEGs.

### 3.3. GO function and KEGG pathway enrichment function analysis

To determine the top 10 genes from the enrichment study of both the upregulated and downregulated top 10 target hub genes, GO analysis was carried out using the TNMplot and Network Analyst tools. These GO findings encompassed categories such as biological processes, molecular activities, cellular components, and KEGG pathways. One particularly relevant criterion used in the analysis was the false discovery rate (FDR). The top 10 target hub genes associated with the cellular component (CC) category included (GO:0005581-collagen trimer), (GO:0090575-RNA polymerase II complex), (GO:0048471-perinuclear region), (GO:0005938-cell cortex), and (GO:0043235-receptor complex). For molecular function (MF), two important protein roles were identified: (GO:0046790-virion binding) and (GO:0001540-beta-amyloid binding). Moreover, the identification of augmented biological processes (BP) in the investigation of DEGs includes (GO:0010629-negative regulation of gene expression), (GO:0042953-lipoprotein transport), (GO:0001774-microglial activation), (GO:0006954-inflammation), (GO:0045429-nitric oxide biosynthesis), (GO:0045087-innate immune response), (GO:0006911-phagocytosis), and (GO:0001666-response to hypoxia). Finally, the KEGG pathways include (hsa04380-Osteoclast differentiation) and (hsa04148-Efferocytosis). Functional details and pathways are listed in Table 3.

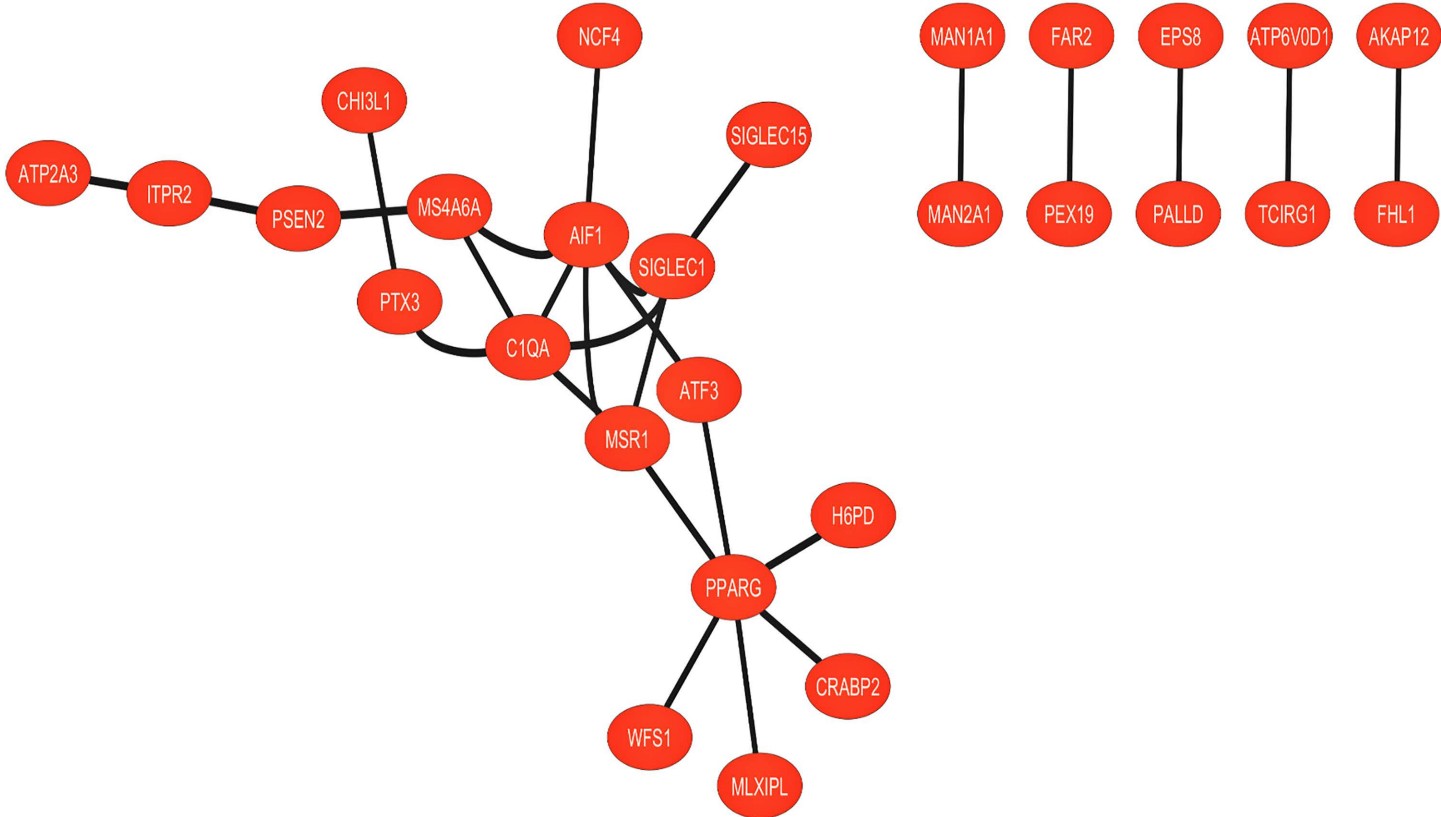

**Fig 2. There were 27 edges and 28 nodes in the network that was built.** The analysis's topological measures revealed robust interconnections within the network, with a mean node value of 1.02 and a mean local cluster coefficient of 0.377. The PPI data of the differentially expressed genes (DEGs) were visualized using Cytoscape software and the STRING database.

### 3.4. Hub genes detection

For both the target data set GSE3624 and GSE25496 the relationship between the hub genes and the nodes was assessed and categorized. Red-colored nodes show strong connections. Nodes with a degree value greater than 6 were considered hub nodes using Cytohubba. Additionally, using CytoHubba, which also computes the eccentricity, betweenness Centrality, and closeness centrality. (see Table 4 and S1 Table in S1 Fig). Notably, with a score of 6, for GSE3624 dataset PPARG showed the highest degree while for GSE25496 dataset AIF1 showed the highest degree score. The hub gene interactions' shortest path network such as, degree score, eccentricity, betweenness Centrality, and closeness centrality are indicated by colors in **Fig 4** and Fig S4 in S1 Fig. The rare neurological disease Polycystic Lipomembranous Osteodysplasia with Sclerosing Leukoencephalopathy (PLOSL), which is associated with lipid metabolism abnormalities, is largely caused by the PPARG gene. Elafibranor and AMG-131) are promising PPARG-targeting medications that aim to restore metabolic balance.

### 3.5. Transcription factors analysis

This work found important protein kinases and transcription factors (TFs) linked to DEGs. These conclusions were drawn from their major contributions to the regulated network's development. A regulatory network was built to develop regulatory complexity comprising kinases, TFs, and the transient proteins they were linked to.

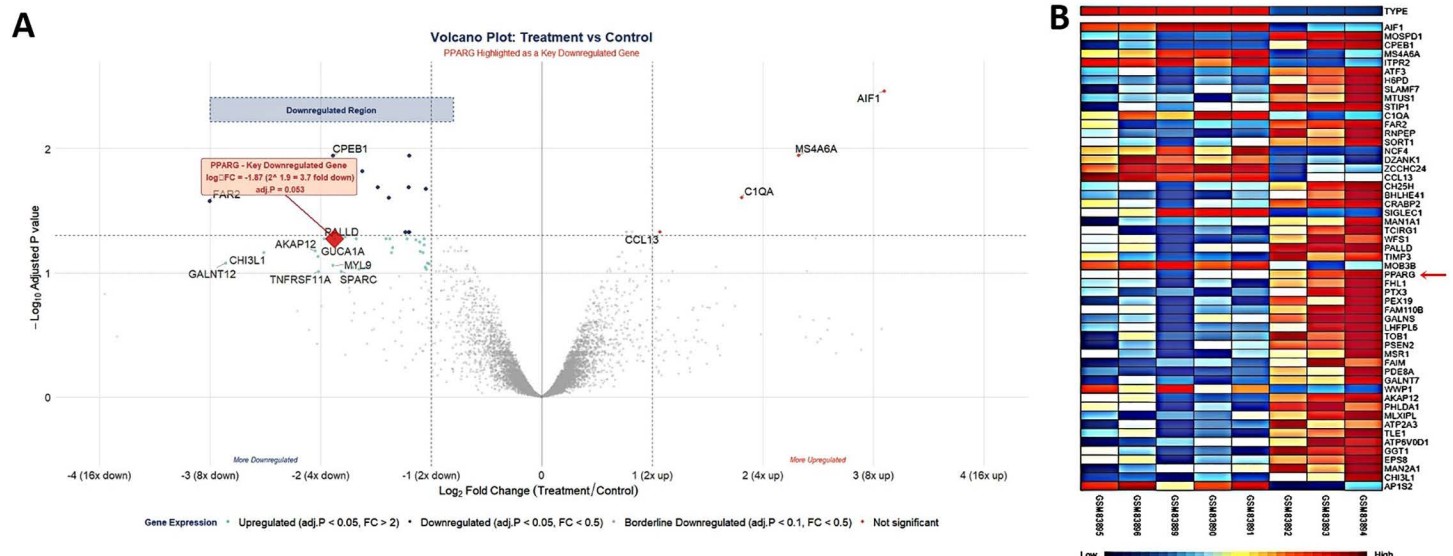

**Fig 3. Displaying the intensity distribution of each gene on a volcano map, with significant downregulated, upregulated genes in the specimens colored in grey based on an adjusted p-value (FDR) < 0.05, and upregulated genes indicated in red and downregulated genes in blue. (B)** Using a heat map to display genes according to their expression levels across all samples. The distribution of expression is shown by the legend, which goes from low (blue) to high (red).

To identify the main transcriptional regulators (TFs), the targeted genes found by ChIP-seq assays (ChEA) must be integrated in the first stage. These TFs were then mapped into networks of PPIs. The transcription factors (TFs) that have been found and the network produced by PPI are shown in Figs 6A and 6B. The main gene transcription factors found in this case are UBTF, NFIC, PPARG, AR, IRF8 and RCOR1, according to the hypergeometric p-value. Moreover, kinase entities, known as PPIs, were discovered and added to the network, indicating that they may play a regulatory role in the enlarged PPI interaction network. The highlighted kinase selections and the corresponding PPI network connections are shown in (Fig 5C–5D). The hypergeometric p-value analysis was also used to determine which dominant kinase was associated with these DEGs. Kinases such as ERK2, MAPK1, CSNK2A1, MAPK14, ERK1, GSK3B, CDK1, MAPK3, CDK4, and HIPK2 have been identified. On the other hand, the identify top 10 hub genes in GSE25496 data set key protein kinases and transcription factors were identified on the base of hypergeometric p-value analysis are display in S5 (A-B) Fig and (C-D). Tables S2 and S3 in S1 Fig the supplemental materials list every TF and kinase discovered and their scores.

### 3.6. Survival analysis of the hub genes

To examine the overall influence on survival of ten major hub genes chosen from both up and down regulated DEGs, we used KMPlot online tool. Among the identified genes investigated, only PTX3 and PPARG had a significant connection with decreased overall survival, with a hazard ratio (HR) of 1.63 and 1.47 respectively, in the group with greater expression levels. Survival analysis employing all hub genes and p-value predictions indicate that these genes significantly influence disease diagnosis and prognosis. These genes may be systematically prioritized in illness detection and therapeutic analysis. Through this KMPlot online tool estimated survival profiles for each hub gene are displayed in **Fig 6**.

### 3.7. Protein-drugs interactions

A drug repository database was consulted to choose medications based on the top 10 hub genes found (Table 5). This investigation used a manual search to find the right drugs for each target. The inquiry panels cover a wide range of

**Table 3. Shows the overview of DEGs distribution in highly significant cellular components, molecular functions, biological processes and KEGG pathways based on FDR significant value.**

| GO_ID | Term description | Rate of False discovery | Matching proteins in the network (labels) |
|---|---|---|---|
| **Cellular components** | | | |
| GO:0005581 | collagen trimer | 0.040243823 | MSR1, C1QA |
| GO:0048471 | perinuclear region of cytoplasm | 0.040433852 | PSEN2, PPARG, AIF1 |
| GO:0090575 | RNA polymerase II transcription factor complex | 0.05278098 | PPARG, ATF3 |
| GO:0005938 | cell cortex | 0.075794764 | PSEN2, ITPR2 |
| GO:0043235 | receptor complex | 0.09392639 | ITPR2, PPARG |
| **Molecular function** | | | |
| GO:0046790 | virion binding | 0.005680907 | PTX3, SIGLEC1 |
| GO:0001540 | beta-amyloid binding | 0.04054124 | MSR1, C1QA |
| **Biological processes** | | | |
| GO:0010629 | negative regulation of gene expression | 0.0089432 | MSR1, PPARG, AIF1 |
| GO:0042953 | lipoprotein transport | 0.009216997 | MSR1, PPARG |
| GO:0001774 | microglial cell activation | 0.013797115 | C1QA, AIF1 |
| GO:0048662 | negative regulation of smooth muscle cell proliferation | 0.015167476 | PPARG, AIF1 |
| GO:0006954 | inflammatory response | 0.016256342 | PTX3, SIGLEC1, AIF1 |
| GO:0045429 | positive regulation of nitric oxide biosynthetic process | 0.021540137 | PTX3, AIF1 |
| GO:0045087 | innate immune response | 0.03279628 | C1QA, PPARG, PTX3 |
| GO:0006911 | phagocytosis, engulfment | 0.052415886 | MSR1, AIF1 |
| GO:0001666 | response to hypoxia | 0.078557149 | PSEN2, ITPR2 |
| **KEGG Pathways** | | | |
| hsa04380 | Osteoclast differentiation | 0.075704469 | ITPR2, PPARG |
| hsa04148 | Efferocytosis | 0.087056657 | C1QA, PPARG |

**Table 4. The STRING protein-protein interaction database determines the top 10 genes found through the DEGs network analysis. The fold changes and corrected values for each gene are also included in the table.**

| Rank | Gene name | Score | logFC | Adjusted P-value | Eccentricity | Betweenness Centrality | Closeness Centrality |
|---|---|---|---|---|---|---|---|
| 1 | PPARG | 6 | −1.8855 | 0.00015343 | 5 | 0.013888889 | 0.391304348 |
| 2 | AIF1 | 6 | 3.1108 | 0.00000034596 | 3 | 0.333333333 | 0.642857143 |
| 3 | C1QA | 5 | 1.8134 | 0.000027186 | 5 | 0 | 0.3 |
| 4 | SIGLEC1 | 4 | 2.4926 | 0.00011945 | 4 | 0 | 0.409090909 |
| 5 | MSR1 | 4 | −1.3591 | 0.00023267 | 4 | 0.222222222 | 0.409090909 |
| 6 | MS4A6A | 3 | 2.3256 | 0.0000044162 | 4 | 0.152777778 | 0.529411765 |
| 7 | PSEN2 | 2 | −0.72774 | 0.00022877 | 3 | 0.319444444 | 0.642857143 |
| 8 | ITPR2 | 2 | 0.99659 | 0.0000067195 | 4 | 0 | 0.5 |
| 9 | PTX3 | 2 | −1.231 | 0.00016209 | 3 | 0.388888889 | 0.5625 |
| 10 | ATF3 | 2 | −1.6259 | 0.0000090115 | 4 | 0.041666667 | 0.45 |

medications, such as FDA-approved medications, experimental medications, and nutraceuticals. A manual screening of medications was done to find interactions with every hub gene-identified protein. For each of them, a total of 11 medications were found. PPARG was linked to two medications, whereas the other hub genes were each connected to a single drug. To guarantee the safety of AMG-131 and Elafibranor in clinical settings, their possible adverse effects should be carefully examined. While Elafibranor, a PPARα/γ dual agonist, has been associated in clinical trials with hepatotoxicity,

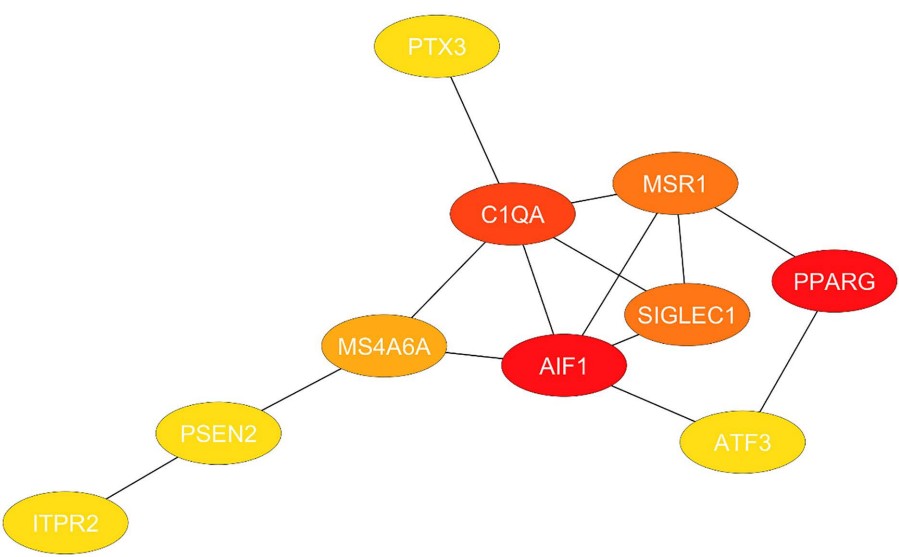

**Fig 4. All hub genes found have their shortest path interactions shown here, with nodes colored according to their degree scores.**

gastrointestinal problems, and possible cardiovascular adverse effects, AMG-131, a PPARG antagonist, may cause metabolic disruptions, insulin resistance, and cardiovascular risks. Analyzing their toxicity, pharmacokinetics, and off-target effects requires a thorough ADMET analysis. The ADMET properties have been inferred via SWISS-ADME server, and the results are shown in Table S4 in S1 Fig.

### 3.8. Molecular Docking of PPARG and its potential drug

A structure-based virtual screening of drug-like compounds was performed in molecular docking to find compounds with a higher affinity for the receptor enzyme active site. PyRx was used to dock inhibitors into the enzyme's active site using the AutoDock Vina plugin for both compounds obtained during the computational screening [42]. Both the best inhibitors were determined to be compound (AMG-131) with the chemical formula [2,4-dichloro-N-(3,5-dichloro-4-quinolin-3-yloxyphenyl)benzenesulfonamide] and (Elafibranor) with the chemical formula [2-[2,6-dimethyl-4-[(E)-3-(4-methyl sulfanyl phenyl)-3-oxoprop-1-enyl]phenoxy]-2-methyl propanoic acid] **Fig 7**. The compounds binding affinity results in −7.4 kcal/mol and −7.2 kcal/mol, respectively. Results were analyzed for control inhibitor, i.e., (Rosiglitazone (S) which inferred −7.5 kcal/mol binding affinity with interactive residues Thr278, Arg371, Gln275, Glu243, Arg316 and His315 as shown in **Fig 8C**. According to **Fig 8**, the chemical was examined for interactions with almost identical active site residues. It was discovered that the compound quinolin-3-yloxyphenyl)benzenesulfonamide] ring and -(4-methyl sulfanyl phenyl)- extended toward the enzyme N-terminus. Alongside, the molecular docking analysis for others pharmacological targets elucidate the binding affinities of five compounds—AMG-131, Elafibranor, Rosiglitazone (S), Samarium Sm-153 Lexidronam, and Clemastine toward three receptors: 1FM6, 2g2b, and 1udw. AMG-131 demonstrates robust binding to all receptors, with the greatest affinity recorded for 1FM6 (−7.4 kcal/mol), followed by 1udw target (−7.1 kcal/mol) and 2g2b target (−6.9 kcal/mol). Elafibranor exhibits substantial binding, especially with 1FM6 (−7.2 kcal/mol), but significantly reduced affinities are seen for 2g2b (−5.8 kcal/mol) and 1udw (−5.6 kcal/mol). The positive control molecule, Rosiglitazone (S), has the highest binding affinity with 1FM6 (−7.5 kcal/mol) and maintains consistent interactions with target 2g2b (−6.1 kcal/mol) and 1udw (−6.1 kcal/mol). Samarium Sm-153 Lexidronam and Clemastine exhibit decreased receptor binding, with affinities between −5.2 kcal/mol and −6.9 kcal/mol respectively as shown in

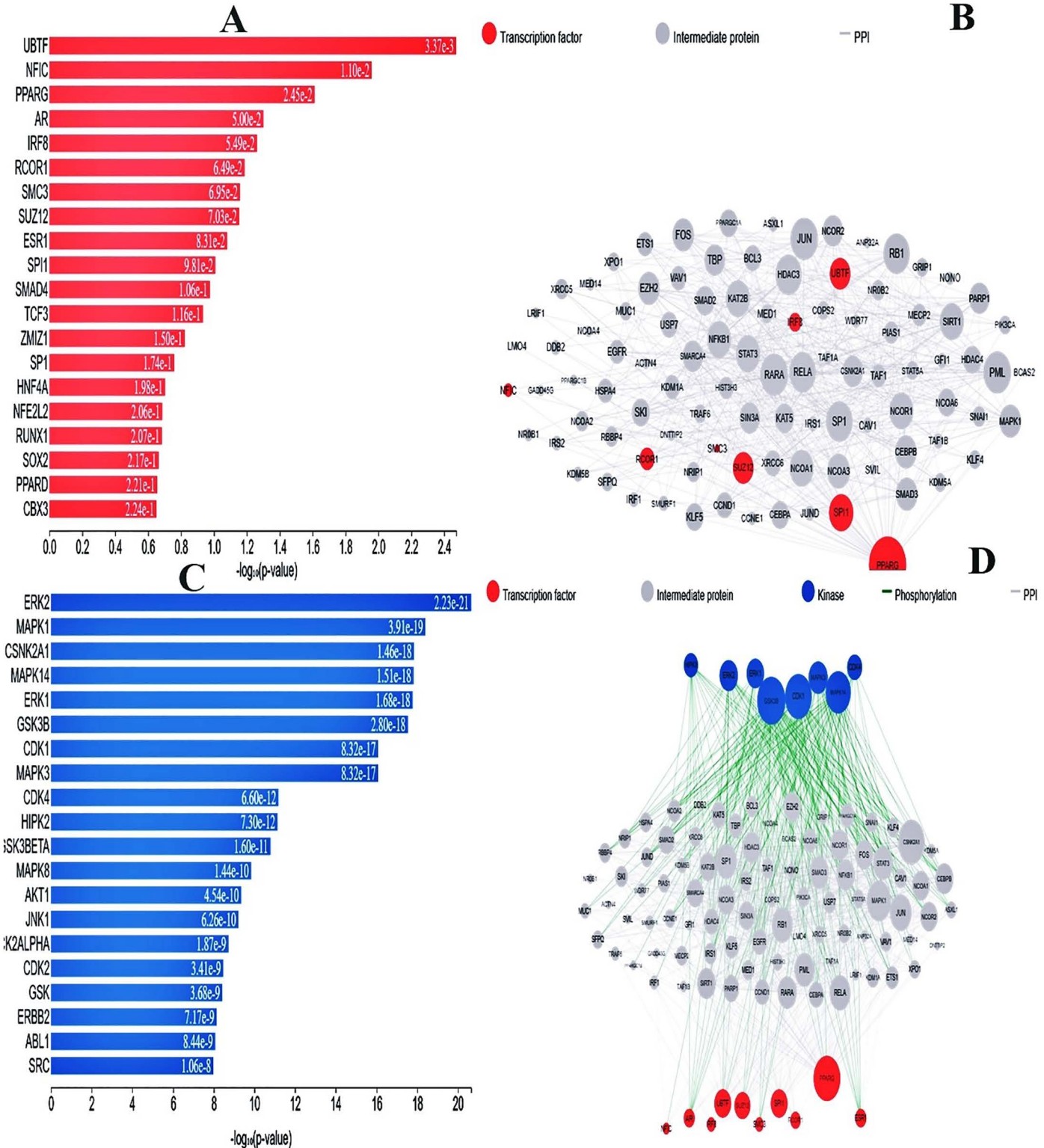

**Fig 5. (A)** A bar graph illustrating the score and hypergeometric likelihood value for every detected element is displayed with a collection of anticipated transcription factors for the DEGs. **(B)** The components of the connected transcripts system and the interacting peptides are shown in a sphere-and-stick model. Gray nodes show the peptides that connect transcription factors, and pink nodes represent transcription factors. The degree of every node in the network is correlated with its size and quantity. **(C)** A bar graph displaying the kinase score (hypergeometric p-value), produced using the X2K Web tool,

presents a ranking of the most likely results. **(D)** A visualization of the PPI networks corresponding to the most often identified kinases in differential gene expression.

Table S5 in S1 Fig. These findings emphasize AMG-131 and Rosiglitazone (S) as viable options, especially for targeting the 1FM6 receptor, while revealing receptor-specific characteristics that might inform future optimization techniques. Alongside, a negative control drug) was run against the PPARG target which resulted in a binding affinity of (−5.2 kcal/mol) against 1FM6. (−4.5 kcal/mol) against 2g2b and (−4.3 kcal/mol) in response to 1udw respectively with low binding interactions. These results confirm the compounds' reliability in preventing the substrate from reaching the active site, preventing the formation of the enzyme-substrate complex and the product. The desired configuration of the molecule involved interactions between the Arg76, Leu28leu Leu118, a residue of the loop region from the central domain against both the compounds inferred strong hydrogen bonds formation and other bonds inside the active domain as shown in **Fig 8A**-**8B** **and** 8C. The distance of hydrogen bonds among residue Arg76 with ligand atom (Fig 8A) is 2.3 Å, 3.4 Å (Fig 8B) along with the other residues whose average distance score results in 3.6 Å shows a high binding affinity.

According to literature in ligand-displacement direct binding assays, the highly potent PPARG ligand 2,4-dichloro-N-(3,5-dichloro-4-(quinolin-3-yloxy)phenyl)benzenesulfonamide has a Ki of 10 nM, which is enough to push rosiglitazone out of the ligand binding pocket [43]. This compound is much more effective than rosiglitazone in cell-based transcriptional activation tests, with an EC50 value of 4 nM and a peak transcriptional activation of reporter genes of about 30%. Cell-based functional tests have demonstrated distinct coregulatory recruitment characteristics of comparison to complete agonists like rosiglitazone [44]. Whereas the compound Elafibranor is an agonist of both peroxisome proliferator-activated receptor-α and peroxisome proliferator-activated receptor-δ. have just been granted FDA expedited clearance after phase 3 studies that showed significant increases in ALP, biochemical response rates, and pruritus alleviation. It can regulate lipid metabolism, enhance glucose homeostasis, and have anti-inflammatory properties [45]. Elafibranor long-term safety and impact on clinical outcomes were assessed in a phase 2 study. Additionally, elafibranor medication was linked to a significant reduction in immunologic and inflammatory indicators, such as circulating levels of the bile acid precursor 7α-hydroxy-4-cholesten-3-one (C4), and immunoglobulin M (IgM) [46].

### 3.9. MD simulation

The Peroxisome proliferator-activated receptor gamma enzyme (PPARG) has several static configurations that have been proposed, either by itself or in conjunction with ligand; it is still necessary to understand the dynamics of the enzyme in an aqueous milieu to properly function. A 200-ns timeline was used to study the dynamics of the protein in combination with the best inhibitor. This makes it possible to develop new effective inhibitors against the enzyme and provides a thorough understanding of the structural changes made to the protein in the presence of ligand. The stability of the protein in complex with ligand was examined during the simulation period using four statistical parameters: Radius of Gyration (Rg), Root Mean Square Fluctuation (RMSF), and RMSD (Fig 9A). The distances between the backbone atoms of overlaid proteins are averaged and expressed as RMSD. The RMSD of the Cα atoms suggests that the enzyme-inhibitor combination is stable throughout the time intervals. The projected mean RMSD value was 2.95 Å for the AMG-131 complex system and 2.93 Å against the Elafibranor complex system, with a maximum RMSD value of 3.7 Å at 100 ns and 4.1 Å at 166 ns, respectively. RMSF values were tracked together with the convergence values for both systems to uncover the structural mobility for every system residue. It is seen that the enzyme residues in confined form remained extremely stable in both systems. The mean root mean square function (RMSF) values for the complex systems were 1.3Å for AMG-131 and Elafibranor complex systems, respectively. Herein, the maximum RMSF value recorded for both systems is 7.5Å at residue 57 and 14.1 Å at the start of the residue. Most residues of the docked protein exhibit RMSF values <2 Å, as **Fig 9B** clearly illustrates. This indicates the PPARG enzyme's extreme stability even further. Finally, to gain a deeper knowledge of the

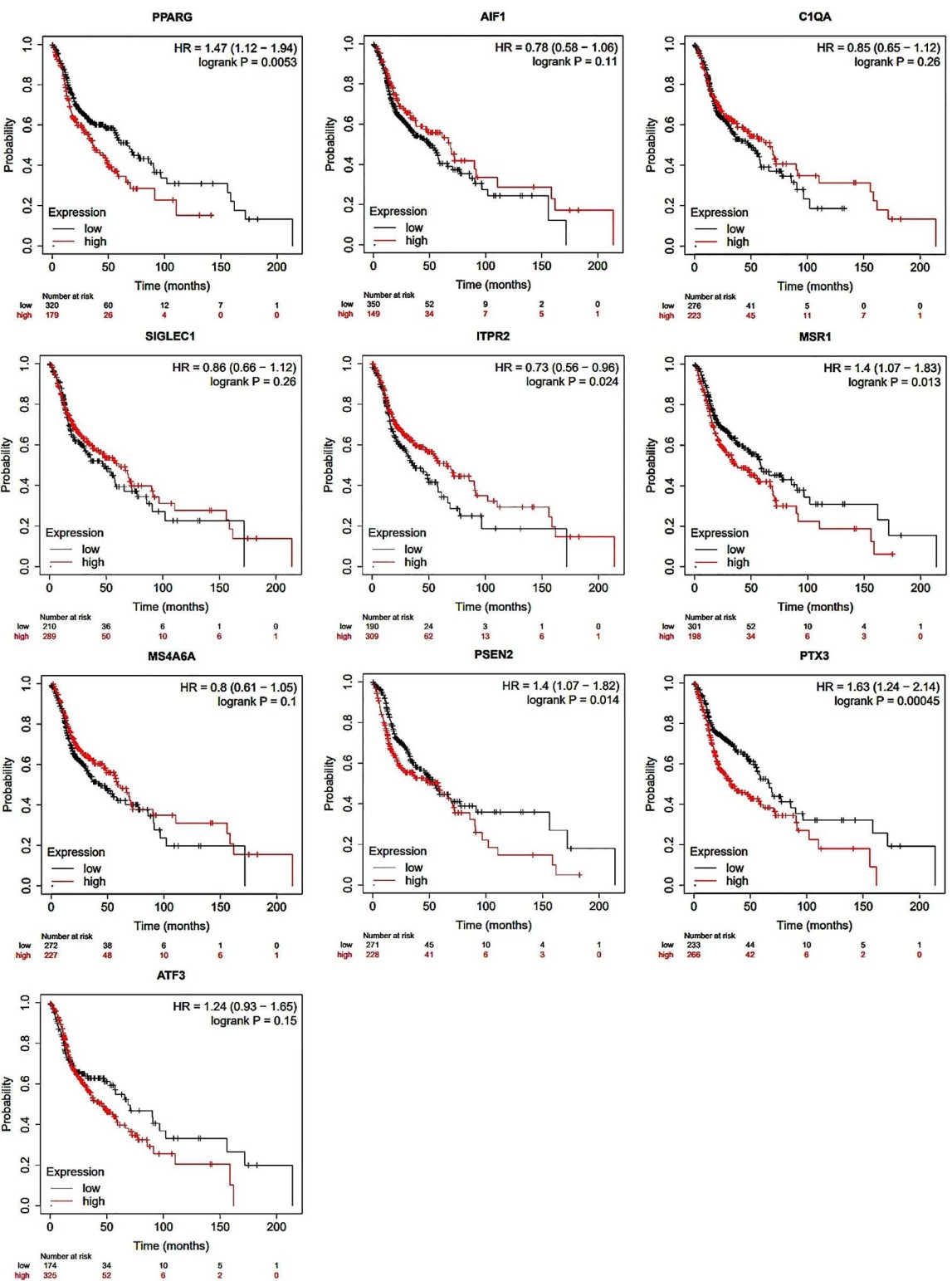

**Fig 6. A lifetime diagram displays every target gene (PPARG, AIF1, C1QA, SIGLEC1, MSR1, MS4A6A, PSEN2, ITPR2, PTX3, and ATF3).** When compared to the low expression of PPARG (black), the red survival curve for PPARG shows a higher survival rate (p < 0.0053). This trend holds true for all the genes based on statistical analysis using the Kaplan-Meier plotter.

**Table 5. Drugs that target protein targets can be identified by analyzing the Drug Bank database.**

| S.NO | Protein | Uniprot ID | Drug name | Drug bank ID | Groups |
|---|---|---|---|---|---|
| 1. | PPARG | P37231<br>Q03181 | AMG-131<br>Elafibranor | DB05490<br>DB05187 | Investigational<br>Investigational |
| 2. | AIF1 | Q9BZX2 | samarium (153Sm) lexidronam | DB05273 | Approved |
| 3. | C1QA | Q9BZX2 | Clemastine | DB00283 | Approved |
| 4. | SIGLEC1 | P23975 | Iobenguane | DB06704 | Approved |
| 5. | MSR1 | Q96SW2 | Thalidomide | DB01041 | Approved |
| 6. | MS4A6A | P40227 | Morphine | DB00295 | Approved |
| 7. | PSEN2 | Q9UHC9 | Cimetidine | DB00501 | Approved |
| 8. | ITPR2 | Q6ZQN7 | Saxagliptin | DB06335 | Approved |
| 9. | PTX3 | P30968 | Leuprolide | DB00007 | Approved |
| 10. | ATF3 | P41143 | Tapentadol | DB06204 | Approved |

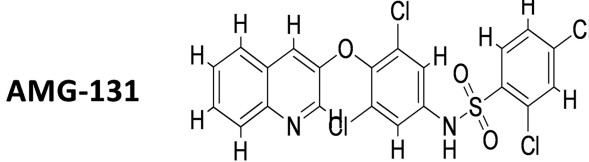

**AMG-131**

2,4-dichloro-*N*-(3,5-dichloro-4-(quinolin-3-yloxy)phenyl)benzenesulfonamide

**Elafibranor**

(*E*)-2-(2,6-dimethyl-4-(3-(4-(methylthio)phenyl)-3-oxoprop-1-en-1-yl)phenoxy)-2-methylpropanoic acid

**Fig 7. Best hit molecules against PPARG protein target.**

systems, the equilibrium conformation of the systems was examined. This was accomplished by figuring out Rg, a measure of how compact a protein structure is. During a simulation run, changes in the Rg measurements affect how compact the protein is with time intervals as shown in **Fig 9C**. Protein amino acids are packed loosely in higher Rg values but tightly in lower Rg values. With a mean value of 19.3 Å for complex AMG-131, a maximum value of 20 Å was recorded at 194 ns and a mean value of 18.9 Å for complex Elafibranor was recorded with a maximum value of 19.5Å at 134 ns. The Rg values for both systems presented here confirm the protein's stability in docked form and are comparable to the RMSD values.

Root mean square deviation (RMSD) and root mean square fluctuation (RMSF) analyses were used in our 100 ns molecular dynamics simulations to quantify the relative stability and binding kinetics of the AMG-131 and Elafibranor complexes (Fig 9). A stiff and well-bound conformation was shown by the AMG-131 complex's lower and more stable backbone and ligand RMSD profiles (Figs 9A, 9B), which were further corroborated by the RMSF plot decreased fluctuations in

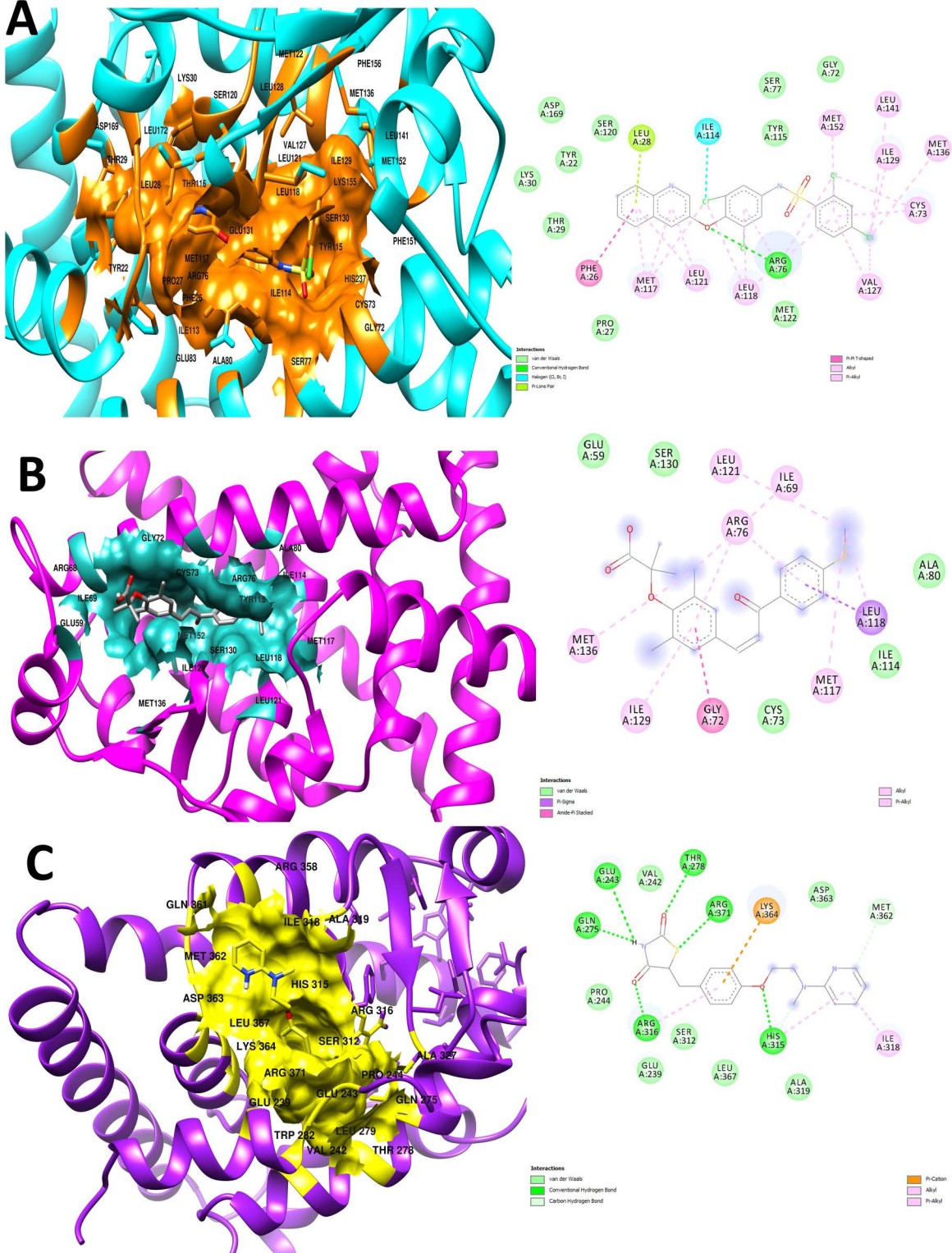

**Fig 8. (A)** Depicting the active pocket with interactive residues at the N-terminal end against both the inhibitors. **(B)** showing the linkages among the receptor and active compounds with benzene rings presenting strong hydrogen bonds, pi-sigma and Alkyl bonds. **(C)** Presenting the control inhibitor against the target protein with binding interactions in 3D and 2D analysis with strong hydrogen bonds and other linkages.

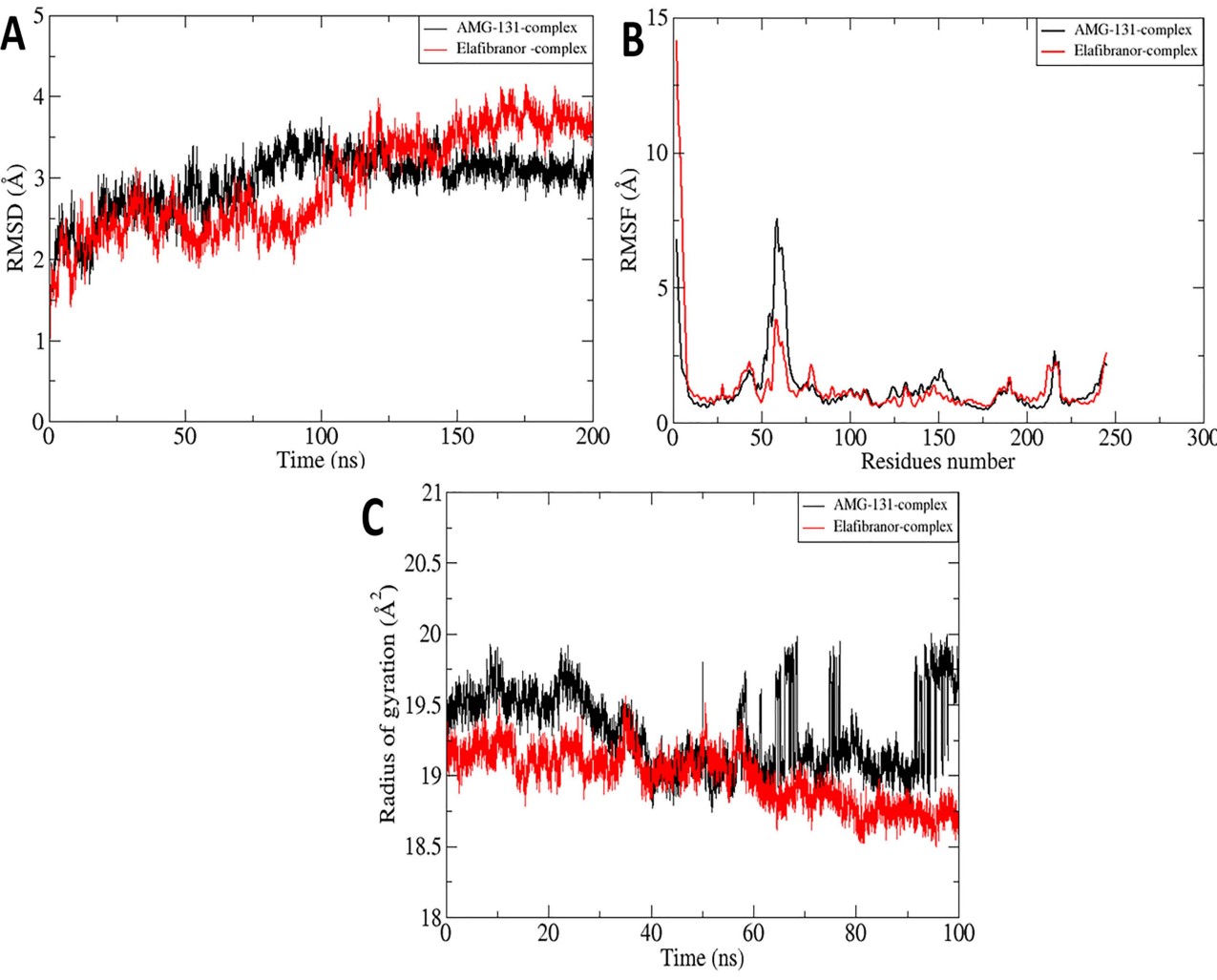

**Fig 9. Depicting the molecular dynamics of complexes obtained during the system biology approach. (A)** Depicting the RMSD of both complex systems for 200 ns simulation intervals. **(B)** The flexibility and thermal effects on residues during the simulation time frame of 200 ns for both complex systems were presented with RMSF analysis, and **(C)** the rigidity of carbon alpha atoms with radius of gyration analysis was illustrated for both complex systems.

binding-site residues (Fig 9C). The ligand's limited angular displacement within the pocket over time confirmed its spatial stability (Fig 10). On the other hand, the Elafibranor complex showed increased residue flexibility (Fig 9C) and higher backbone and ligand RMSD values with larger variability (Figs 9A, 9B), which is in line with its notable angular reorientations seen along the trajectory (Fig 10). Together, these findings imply that, in contrast to the more dynamic and mobile Elafibranor-bound state, AMG-131 forms a more stable and conformationally restricted complex.

## 4. Discussion

Comprehensive investigation of molecular changes linked to disease has been made possible by developments in digital and genomic technologies, such as SNP analysis, GWAS, and microarray-based expression profiling. Through multi-level omics, such as transcriptomics, proteomics, metabolomics, and genomes, these techniques facilitate the identification of biomarkers and enhance comprehension of disease causes. [16,47]. In this study, protein–protein interaction (PPI)

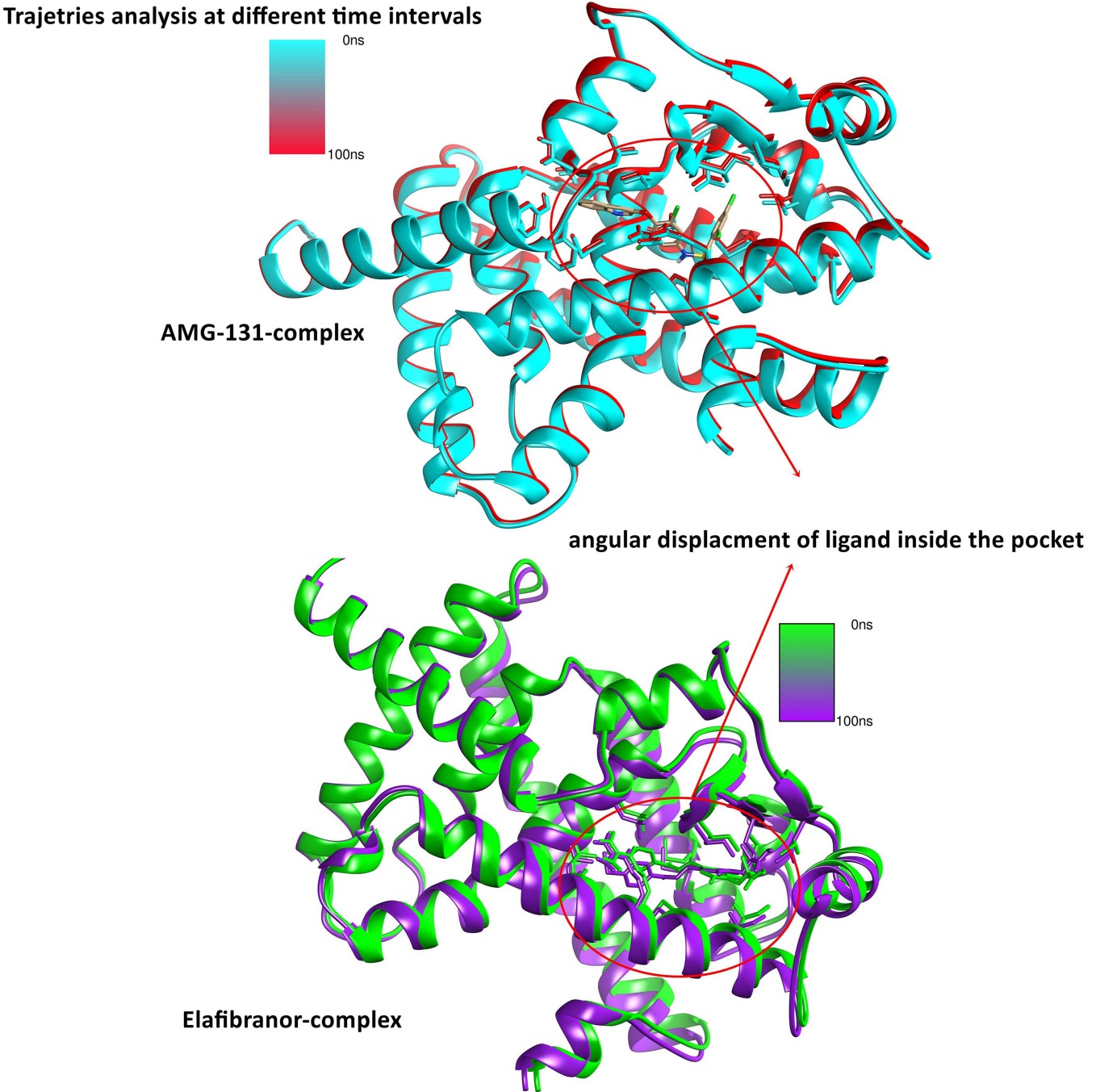

**Trajetries analysis at different time intervals**

AMG-131-complex

angular displacment of ligand inside the pocket

Elafibranor-complex

**Fig 10. Depicting the complex systems after 0 ns and 100 ns time intervals with minor angular displacement of ligand inside the pocket throughout the simulations.**

networks were constructed to highlight important regulators after differentially expressed genes (DEGs) were identified using microarray datasets (GSE3624 and GSE25496). Although instructive, both datasets' small sample sizes lower statistical power and raise the possibility of false positives or false negatives, highlighting the need for careful interpretation

and further validation with bigger cohorts. PPI network analysis is frequently used to understand metabolic processes, find possible treatment targets, and clarify disease causes [48]. The structural relationships between several proteins greatly influence numerous biological processes, which might differ between healthy and pathological states [48,49]. The identification of genes that are expressed differently in disease situations than in healthy ones, microarray gene expression data analysis helps identify targets for new medication development. Molecular network connections improve the precision and consistency of identifying illness-associated biomarkers [50]. Prior studies have indicated the significance of these analyses in forecasting central nodes and their vital roles in a range of illnesses [51,52].

Regarding PLOSL patients, the prognosis for those in the early stages is far better than that for those in the later stages. Early-stage illness patients frequently experience excellent cure rates while receiving chemotherapy, radiation therapy, surgery, or a combination of these therapies. On the other hand, those who have advanced-stage PLOSL have more difficulties, mainly because the illness is aggressive, and a solution is sometimes unattainable [53]. This study used microarray data to find DEGs in PLOSL. Ten hub genes were shown to be potential contributors to PLOSL disease by PPI network and enrichment analysis, with PPARG appearing as the most promising candidate. Prior research demonstrates that PPARG is involved in processes related to the development of PLOSL, such as inflammation, lipid regulation, and cell-proliferative activity [54]. It has been demonstrated that PPARG has a major influence on PLOSL formation [55]. PPARG's role in disease-related processes, such as increased cell proliferation, made it a viable target for intervention [56]. In a recent study, it has been revealed that microglial PPARG signaling promotes the wound healing process by modulating transcriptional events, such as limiting chronic microglial activation. This explains the causal role of PPARG in downstream functional decline and in regulating neuroinflammation [57]. Additionally, preclinical findings suggest the suppression of NF-κB/inflammasome signaling pathways and microglia shift to anti-inflammatory phenotypes, illustrating the PPARG as a neuroinflammation hub [58]. The roles of the other identified targets are critical to the evolution of PLOSL. Their relevance was highlighted by enrichment analysis, subnetwork building, and the incorporation of all hub genes inside these subnetworks. A review of KEGG pathways, molecular activities, cellular constituents, and biological processes clarified the functions of the genes in several interrelated pathways. Identifying transcription factors and their roles within the expanded network enabled determining their regulatory functions. Previous studies have noted the importance of ECM–receptor interaction and Notch signaling pathways in PLOSL, also included in our KEGG pathway analysis. Additionally, our research indicates significant alterations in immune-related pathways in PLOSL patients [53].

However, several other hub genes that have been found, like AIF1 and C1QA, also seem to have biological significance. C1QA is essential for immune-mediated clearance and synaptic remodeling, whereas AIF1 is a microglial activation marker associated with phagocytic regulation and neuroinflammation. When taken as a whole, these genes demonstrate the significant role that dysregulated immunological signaling, microglial dysfunction, and compromised efferocytosis play in PLOSL. Their dispersion among biological processes, molecular activities, and KEGG pathways provides more evidence that the disease is complex rather than dependent on a single dominant driver.

Molecular dynamics (MD) simulations, docking, and structural modeling were used concurrently to evaluate PPARG as a possible therapeutic target. Docking was used to screen potential FDA-approved drugs, and then MD simulations were used to assess receptor–ligand stability. Although AMG-131 and Elafibranor showed good in-silico binding characteristics, these results are still in the early stages. Crucially, AMG-131 is still an early-stage research chemical without therapeutic licensure, and Elafibranor's clinical development has been halted due to unsatisfactory trials. As a result, none of the discovered compounds are currently thought to be clinically feasible for PLOSL, and thorough in-vitro and in-vivo validation of computational predictions is necessary. Even though our computational investigations showed that AMG-131 and Elafibranor have favorable binding interactions with PPARG, it is crucial to remember that neither drug is now therapeutically viable. The development of Elafibranor, a dual PPARα/γ agonist, was stopped after it failed to meet primary endpoints in clinical trials. There is currently no authorized therapeutic indication for AMG-131, making it an early-stage investigative chemical. Therefore, before any evaluation of therapeutic relevance, the results given here should be carefully understood as preliminary in-silico discoveries that require considerable preclinical and experimental validation.

## 5. Conclusions

To find possible molecular contributors and prospective drugs related to PLOSL, this study used differential expression analysis, PPI network design, functional enrichment, and structure-based computational modeling. PPARG and many other hub genes, such as AIF1 and C1QA, were identified by the analysis as potential important regulators of immune-related pathways and microglial dysfunction. The molecular processes that may explain PLOSL disease are first revealed by these results. AMG-131 and Elafibranor have preferential binding to PPARG, according to computational docking and molecular dynamics simulations; nonetheless, these findings should be interpreted cautiously. With AMG-131 still in the experimental stage and Elafibranor no longer in clinical research, both substances are currently unvalidated in the context of PLOSL. Consequently, treatment preparedness is not indicated by the interaction predictions shown here. All things considered, this work provides an initial framework for researching PLOSL that generates hypotheses. The results are exploratory and computational in nature; further in-vitro, in-vivo, and clinical validation will be required to ascertain the actual biological significance of the identified genes and chemicals. To get these candidates closer to any possible therapeutic use, more extensive datasets and experimental validation must be included in future studies.

Finding new pharmacological targets that contribute to the onset and development of PLOSL is crucial, in addition to the biomarkers already used for the condition, which contain mutations that result in therapy resistance. The statistical strength of the results is limited because the datasets employed in this investigation were based on very small sample sizes. Even though our computational analysis suggested PPARG and other hub genes as possible targets for treatment, these findings should be considered preliminary. To establish the biological significance of the suggested indicators and to further assess the therapeutic potential of AMG-131 and Elafibranor, future studies utilizing larger patient cohorts and rigorous experimental validation in both in vitro and in vivo models are crucial.

This research aims to identify novel biomarkers linked to PLOSL that may be used to control the condition's occurrence worldwide. The two therapeutic drugs, AMG-131 and Elafibranor, were chosen for binding study against the target gene and verified using molecular docking methods based on encouraging scores along with the control inhibtor. This study will help us better understand how drugs resist their effects and pave the way for developing novel therapeutics and diagnostics. The drug's mechanical behavior and mechanism of action toward the target biomarker gene PPARG were revealed by MD investigations. The drugs under study bind tightly to the active site and maintain this binding throughout the simulation period. This strategy for biomarker discovery delivers more accuracy and cost-effectiveness than conventional experimental approaches by utilizing artificial intelligence and computational methodologies, eventually enhancing PLOSL research. The predicted compounds can be experimentally tested using different in vitro approaches; binding assays, enzymatic assays, cell-based functional assays, and phenotypic screening (Jennings, 2015). The drug can be further validated using in vivo methods in animal models such as like rats and mice.

## Supporting information

**S1 Fig. Prior to normalization, the box plot, density plot, and mean expression plot are shown in (A), (B), and (C); following normalization, the same plots are shown in (D), (E), and (F). Using the Network Analyst Tool, variance-stabilizing normalization successfully removed noise from the data. The next methods used to analyze the data were Principal Component Analysis and the Quality Control Examination.**
(DOCX)

## Author contributions

**Conceptualization:** Bayan T. Bokhari, Alaa Abdulaziz Eisa.

**Data curation:** Bayan T. Bokhari, Alaa M. Saleh, Hashim M. Aljohani, Wejdan Hussain Owaydhah, Mohammad Ahmad Alobaidy, Naief Dahran, Hind M Naffadi, Alaa Abdulaziz Eisa.

**Formal analysis:** Bayan T. Bokhari, Alaa M. Saleh, Hashim M. Aljohani, Wejdan Hussain Owaydhah, Mohammad Ahmad Alobaidy, Alaa Abdulaziz Eisa.

**Funding acquisition:** Bayan T. Bokhari, Alaa M. Saleh, Alaa Abdulaziz Eisa.

**Investigation:** Bayan T. Bokhari, Alaa M. Saleh, Hashim M. Aljohani, Wejdan Hussain Owaydhah, Mohammad Ahmad Alobaidy, Naief Dahran, Hind M Naffadi, Alaa Abdulaziz Eisa.

**Methodology:** Bayan T. Bokhari, Alaa M. Saleh, Hashim M. Aljohani, Wejdan Hussain Owaydhah, Mohammad Ahmad Alobaidy, Naief Dahran, Hind M Naffadi, Alaa Abdulaziz Eisa.

**Project administration:** Bayan T. Bokhari, Alaa M. Saleh, Alaa Abdulaziz Eisa.

**Resources:** Bayan T. Bokhari, Hashim M. Aljohani, Alaa Abdulaziz Eisa.

**Software:** Bayan T. Bokhari, Hashim M. Aljohani, Alaa Abdulaziz Eisa.

**Supervision:** Bayan T. Bokhari, Alaa Abdulaziz Eisa.

**Validation:** Bayan T. Bokhari, Hashim M. Aljohani, Mohammad Ahmad Alobaidy, Alaa Abdulaziz Eisa.

**Visualization:** Bayan T. Bokhari, Naief Dahran, Alaa Abdulaziz Eisa.

**Writing – original draft:** Bayan T. Bokhari, Alaa M. Saleh, Hashim M. Aljohani, Wejdan Hussain Owaydhah, Mohammad Ahmad Alobaidy, Naief Dahran, Hind M Naffadi, Alaa Abdulaziz Eisa.

**Writing – review & editing:** Bayan T. Bokhari, Hashim M. Aljohani, Naief Dahran, Hind M Naffadi, Alaa Abdulaziz Eisa.

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
