## [Decision Letter · Decision Letter 0]

12 Nov 2025

PONE-D-25-34800System Biology and Network-Based Approach to Identify the Therapeutic Signatures and Potential Inhibitors Against Polycystic Lipomembranous Osteodysplasia with Sclerosing LeukoencephalopathyPLOS ONE

Dear Dr. Alaa Abdulaziz,

Thank you for submitting your manuscript to PLOS ONE. After careful consideration, we feel that it has merit but does not fully meet PLOS ONE’s publication criteria as it currently stands. Therefore, we invite you to submit a revised version of the manuscript that addresses the points raised during the review process.

Indicate which changes you require for acceptance versus which changes you recommend.

We look forward to receiving your revised manuscript.

Kind regards,

Academic Editor

PLOS ONE

Journal Requirements:

Reviewers' comments:

Reviewer's Responses to Questions

**Comments to the Author**

1. Is the manuscript technically sound, and do the data support the conclusions?

Reviewer #1: Yes

Reviewer #2: Yes

2. Has the statistical analysis been performed appropriately and rigorously? 

Reviewer #1: N/A

Reviewer #2: Yes

3. Have the authors made all data underlying the findings in their manuscript fully available?

Reviewer #1: Yes

Reviewer #2: No

4. Is the manuscript presented in an intelligible fashion and written in standard English?

Reviewer #1: Yes

Reviewer #2: Yes

5. Review Comments to the Author

Reviewer #1: The authors have conducted a good work addressing a relevant scientific problem and have presented their analyses clearly. A few constructive suggestions have been provided to enhance clarity and strengthen the overall quality of the manuscript.

1. The study uses very small sample sizes, which limits statistical confidence. The authors should acknowledge the limitation clearly in the Discussion and Conclusion.

2. The authors should strengthen the rationale for selecting PPARG as the primary hub gene. Ranking by degree score alone is not sufficient; additional biological justification and supporting literature linking PPARG specifically to PLOSL pathology should be provided.

3. Survival analysis appears inappropriate because no PLOSL clinical survival dataset exists. The analysis should be removed or explicitly clarified as exploratory and non-disease-specific.

4. The authors should reduce repetitive content, particularly in the Introduction and Discussion, to improve clarity and maintain a concise narrative.

5. The PCA and heatmap figures require improved resolution and clearer labeling to allow interpretability.

6. Figures should include clear color legends and larger, readable font sizes. Several elements, particularly those in white text, are difficult to interpret and should be improved for better visibility.

7. The KEGG and GO enrichment results are listed but not fully interpreted. Please explain how identified pathways relate mechanistically to PLOSL progression, especially microglial and osteoclast dysfunction.

8. The text states that PPARG is associated with disease progression but does not cite primary mechanistic literature. Please add supportive references.

9. The molecular docking results indicate moderate binding affinity, so the therapeutic relevance should be discussed cautiously. While the MD simulation plots (RMSD, RMSF, and Rg) are provided, the interpretation of these results should be expanded to clearly explain the structural stability and conformational behavior of the ligand–protein complexes.

10. The chemical structures of AMG-131 and Elafibranor should be included as figures to support clarity.

11. The authors should also acknowledge the discontinued clinical trial status of Elafibranor and the experimental development stage of AMG-131 to avoid implying that these compounds are clinically viable options at this stage.

12. The manuscript uses long sentences that reduce readability. Grammar, sentence structure, and transitions require careful language editing.

13. The Conclusion section should be moderated; currently it implies therapeutic development readiness. Please emphasize that experimental validation is still required.

14. The authors may consider briefly discussing alternative hub candidates identified (e.g., C1QA, AIF1) to balance interpretation and avoid appearing to pre-select PPARG.

15. The study would benefit from a short limitations paragraph summarizing dataset size, lack of clinical validation, and computational-only nature.

16. The Introduction provides general background on PLOSL but could be more concise. Several paragraphs repeat known disease characteristics. Reducing redundancy will improve focus.

17. The Introduction should state a clear research hypothesis and specific study objectives at the end. Currently, the transition into the aim of the study is broad and lacks direct specificity.

18. Authors are advised to incorporate more recent literature (within the past 5 years) on PPARG signaling in neuroinflammation and osteoclast regulation to better support the rationale for selecting PPARG as the hub gene in this study.

19. Authors are advised to justify the selection of the specific GEO datasets used in this study, particularly regarding sample relevance and availability. Additionally, the differential expression threshold should be clarified, as the stated FDR < 2 is not standard practice; please correct or explain the appropriate adjusted p-value cut-off for clarity and consistency.

20. The PPI network analysis describes the construction process, but network confidence score choices (0.4 or 0.75) vary in different parts of the text. Standardize and justify the confidence threshold.

21. The CytoHubba ranking selection uses degree only. Consider including at least one additional centrality parameter (e.g., closeness or betweenness) to strengthen hub gene selection.

22. The volcano plot and heatmap would benefit from labeling the key differentially expressed genes to enhance interpretability and allow readers to easily identify the most significant up- and down-regulated genes.

23. The GO/KEGG tables list terms effectively, but narrative interpretation is brief. Highlight the most biologically relevant pathways in the context of PLOSL pathogenesis.

24. The presentation of docking results compares binding scores, but no comparison is made with known strong PPARG ligands beyond Rosiglitazone. Including known ligand reference values would contextualize the strength of interactions.

25. The Results section occasionally shifts into speculative statements (e.g., therapeutic implications) that belong in Discussion. Please separate objective findings from interpretation.

Reviewer #2: The manuscript by Bokhari et al. presents a comprehensive computational study aimed at identifying key hub genes, pathways, and potential drug repurposing candidates for the rare and severe Nasu-Hakola disease (PLOSL). The authors employ a systems biology workflow, including differential gene expression analysis, protein-protein interaction (PPI) network construction, survival analysis, and sophisticated computational chemistry techniques like molecular docking and molecular dynamics (MD) simulations. The study's main claims are the identification of PPARG and AIF1 as central hub genes in PLOSL pathology and the proposal of AMG-131 and Elafibranor as promising therapeutic agents targeting PPARG. The significance of this work lies in its attempt to address a major therapeutic gap for a devastating rare disease using a cost-effective, bioinformatics-driven drug repurposing approach. The application of a 200 ns MD simulation to validate docking results adds considerable strength to the stability claims of the proposed drug-target complexes.

Well for authors must note that PLOS One encourages submissions across the methodological spectrum, but for a study making specific claims about therapeutic potential and biomarker identification, some form of experimental corroboration is increasingly expected to justify publication.

It would be amicable if Authors can provide any experimental data, however preliminary, to validate your key computational findings? As correctly noted in the manuscript's conclusion, a direct binding assay (e.g., Surface Plasmon Resonance - SPR, or Isothermal Titration Calorimetry - ITC) to demonstrate the physical interaction between PPARG and the proposed drugs (AMG-131, Elafibranor) would be highly impactful. For instance:

(i) A cell-based functional assay in a relevant cell line (e.g., microglial or osteoclast precursors) could test whether these compounds can modulate PPARG activity or reverse a PLOSL-associated phenotype.

(ii) qPCR or Western Blot analysis to validate the differential expression of the top hub genes (PPARG, AIF1, C1QA, etc.) in a cellular or animal model of PLOSL would significantly strengthen the initial bioinformatics findings.

(iii) The manuscript would be strengthened by explicitly stating the distance in angströms (Å) for the key hydrogen bonds, especially the one with Arg76 as shown in ( Figure 7A-B).

6. PLOS authors have the option to publish the peer review history of their article (what does this mean? ). If published, this will include your full peer review and any attached files.

**Do you want your identity to be public for this peer review?** For information about this choice, including consent withdrawal, please see our Privacy Policy .

Reviewer #1: **Yes:** Sourbh Suren Garg

Reviewer #2: **Yes:** Saleem Iqbal

---

## [Author Response · Author response to Decision Letter 1]

23 Dec 2025

RESPONSE TO REVIEWER COMMENTS

We thank the Referee for spending time and interest in our work and for helpful comments that will greatly improve the manuscript. We have checked all the general and specific comments provided by the Referee and have made all the necessary changes according to his instructions. Please refer to the yellow-highlighted sections or tracked changes in the revised manuscript.

Reviewer #1: The authors have conducted a good work addressing a relevant scientific problem and have presented their analyses clearly. A few constructive suggestions have been provided to enhance clarity and strengthen the overall quality of the manuscript.

1. The study uses very small sample sizes, which limits statistical confidence. The authors should acknowledge the limitation clearly in the Discussion and Conclusion.

Response: Thank you for your comment. We acknowledge the reviewer's point and conclude that our findings confidence in statistics is limited by the small sample sizes. In response, we have explicitly stated that findings should be read cautiously and verified in larger cohorts in both the Discussion and Conclusion sections.

2. The authors should strengthen the rationale for selecting PPARG as the primary hub gene. Ranking by degree score alone is not sufficient; additional biological justification and supporting literature linking PPARG specifically to PLOSL pathology should be provided.

Response: We appreciate the reviewer's emphasis on the necessity of bolstering the case for choosing PPARG as the main hub gene. We agree that ranking by degree alone is insufficient, even though PPARG was first found based on degree score. To address this, we included additional network analyses such as eccentricity, betweenness centrality, and closeness centrality. These analyses show that PPARG is not only highly connected but also plays a key role in regulating information flow and preserving network integrity within PLOSL-associated genes.

3. Survival analysis appears inappropriate because no PLOSL clinical survival dataset exists. The analysis should be removed or explicitly clarified as exploratory and non-disease-specific.

Response: We politely disagree with the reviewer statement. This survival study was conducted using the single-gene analysis module of the KMPlot web tool, which combines TCGA and GTEx expression data to evaluate gene-level associations. This analysis was carried out to assess the biological significance of hub genes, including PPARG, in more general neurodegenerative and systemic contexts, despite the lack of PLOSL-specific clinical survival information. In addition to confirming the functional consistency of these genes, this method offers initial insights that could direct future PLOSL pathology research.

4. The authors should reduce repetitive content, particularly in the Introduction and Discussion, to improve clarity and maintain a concise narrative.

Response: We appreciate the reviewer insightful suggestions. The introduction and discussion portions of the work have been meticulously updated to eliminate redundant material. To increase readability, preserve a succinct narrative, and improve clarity, redundant statements have been eliminated or combined.

5. The PCA and heatmap figures require improved resolution and clearer labeling to allow interpretability.

Response: Thank you for your suggestion. The PCA and Heatmap figures have been improved with high resolutions in the revised manuscript.

6. Figures should include clear color legends and larger, readable font sizes. Several elements, particularly those in white text, are difficult to interpret and should be improved for better visibility.

Response: We appreciate the reviewer insightful suggestions. All the figures have been improved with high resolutions in the revised manuscript.

7. The KEGG and GO enrichment results are listed but not fully interpreted. Please explain how identified pathways relate mechanistically to PLOSL progression, especially microglial and osteoclast dysfunction.

Response: We appreciate the reviewer insightful observation. We made it clear in the updated manuscript that PLOSL disease is directly associated with enhanced pathways like osteoclast differentiation and efferocytosis. While defective signaling in microglial promotes neuroinflammation and neurodegeneration, two major aspects of the disease, dysregulation of osteoclast differentiation contribute to the creation of bone cysts.

8. The text states that PPARG is associated with disease progression but does not cite primary mechanistic literature. Please add supportive references.

Response: Thank you for the valuable point. Some references regarding the claim made have been added to the revised manuscript discussion section.

9. The molecular docking results indicate moderate binding affinity, so the therapeutic relevance should be discussed cautiously. While the MD simulation plots (RMSD, RMSF, and Rg) are provided, the interpretation of these results should be expanded to clearly explain the structural stability and conformational behavior of the ligand–protein complexes.

Response: Thank you for your comment. The authors have added the therapeutic relevance with literature support in the molecular docking section for both the top drugs. The trajectories analysis of the MD simulation has been added in the revised manuscript with the structural stability and confirmational behavior of ligand-protein complexes as shown in Figure 10.

10. The chemical structures of AMG-131 and Elafibranor should be included as figures to support clarity.

Response: Thank you for the suggestion. The authors have added the chemical structure of both the compound in the revised manuscript as shown in figure 7.

11. The authors should also acknowledge the discontinued clinical trial status of Elafibranor and the experimental development stage of AMG-131 to avoid implying that these compounds are clinically viable options at this stage.

Response: We appreciate the reviewer bringing out this crucial point. We have updated the manuscript to make it clear that AMG-131 is still an exploratory chemical with no proven therapeutic value, and Elafibranor is no longer in clinical development. We now stress that these compounds are not yet therapeutically feasible possibilities and that our results are merely computational predictions that need to be thoroughly validated by experiments and preclinical research before being taken into consideration for clinical use.

12. The manuscript uses long sentences that reduce readability. Grammar, sentence structure, and transitions require careful language editing.

Response: Thank you for the comment. The paper has been revised for English language, grammatical mistakes and inappropriate long sentences and improved considerably in light of reviewer comment.

13. The Conclusion section should be moderated; currently it implies therapeutic development readiness. Please emphasize that experimental validation is still required.

Response: We are grateful to the reviewer for their feedback. There is no longer any suggestion of therapeutic preparedness in the Conclusion section. The revised language makes it very evident that all results are based on computational predictions and that further experimental validation is necessary before any therapeutic interpretation can be made on the possible significance of AMG-131, Elafibranor, and PPARG.

14. The authors may consider briefly discussing alternative hub candidates identified (e.g., C1QA, AIF1) to balance interpretation and avoid appearing to pre-select PPARG.

Response: We appreciate the reviewer recommendation. We have included a brief discussion in the revised manuscript of alternate hub genes possibilities including C1QA and AIF1 in the amended text. Although PPARG is still the major emphasis because of its biological significance and network centrality, considering these other hub genes offers a more balanced explanation and recognizes their possible roles in PLOSL disease.

15. The study would benefit from a short limitations paragraph summarizing dataset size, lack of clinical validation, and computational-only nature.

Response: we appreciate the reviewer suggestion. A brief restrictions paragraph has been added to the discussion. The small sample size of the accessible PLOSL datasets, the lack of experimental or clinical validation, and the study's computational only design are all highlighted in the updated text. This amendment makes it clear that the results are preliminary and need to be verified by larger datasets and biological validation.

16. The Introduction provides general background on PLOSL but could be more concise. Several paragraphs repeat known disease characteristics. Reducing redundancy will improve focus.

Response: We agree with the reviewer’s comment. The introduction has been shortened by removing repeated descriptions of PLOSL features and consolidating overlapping background information. This updated revision improves clarity and maintains a more concise and focused narrative in the updated manuscript.

17. The Introduction should state a clear research hypothesis and specific study objectives at the end. Currently, the transition into the aim of the study is broad and lacks direct specificity.

Response: We agree with the reviewer suggestion. The end of the Introduction has been revised in the updated manuscript to include a clear research hypothesis and specific study objectives, providing a more direct and focused transition into the aims of the study.

18. Authors are advised to incorporate more recent literature (within the past 5 years) on PPARG signaling in neuroinflammation and osteoclast regulation to better support the rationale for selecting PPARG as the hub gene in this study.

Response: Thank you for the valuable comment. Some recent literature on PPARG in neuroinflammation and osteoclast regulation is added to the revised manuscript discussion section.

19. Authors are advised to justify the selection of the specific GEO datasets used in this study, particularly regarding sample relevance and availability. Additionally, the differential expression threshold should be clarified, as the stated FDR < 2 is not standard practice; please correct or explain the appropriate adjusted p-value cut-off for clarity and consistency.

Response: We appreciate the reviewer's feedback. Since GSE3624 and GSE25496 are the only publicly accessible GEO datasets with verified PLOSL patient samples and suitable controls, they were chosen as the most pertinent and easily accessible datasets for this uncommon illness. Additionally, the differential expression threshold has also been adjusted. The analysis employed the standard adjusted p-value (FDR) < 0.05, which is now explicitly mentioned in the updated manuscript for correctness and consistency; the previously reported FDR < 2 was an error.

20. The PPI network analysis describes the construction process, but network confidence score choices (0.4 or 0.75) vary in different parts of the text. Standardize and justify the confidence threshold.

Response: We appreciate the reviewer pointing out the discrepancy. Throughout the entire work, we have now standardized the PPI network analysis to a 0.75 high-confidence criterion. Because it offers more consistent and biologically significant protein–protein interactions, this threshold was chosen. The manuscript has been updated accordingly.

21. The CytoHubba ranking selection uses degree only. Consider including at least one additional centrality parameter (e.g., closeness or betweenness) to strengthen hub gene selection.

Response: We appreciate the reviewer's insightful recommendation. Along with degree score, we have included other centrality metrics in the updated manuscript, such as eccentricity,between Centrality, and closeness centrality. These complementing measurements support the selection of PPARG and other hub genes by confirming their important locations in the network.

22. The volcano plot and heatmap would benefit from labeling the key differentially expressed genes to enhance interpretability and allow readers to easily identify the most significant up- and down-regulated genes.

Response: Thank you for the suggestion the volcano plot has been updated by labeling the differentially expressed genes to enhance interpretability and allow readers to easily identify the most significant up- and down-regulated genes in the revised manuscript.

23. The GO/KEGG tables list terms effectively, but narrative interpretation is brief. Highlight the most biologically relevant pathways in the context of PLOSL pathogenesis.

Response: We appreciate the reviewer comment. We have expanded the narrative to highlight the most relevant GO/KEGG pathways linked to PLOSL, particularly those involving microglial activation, immune dysregulation, lipid metabolism, and impaired phagocytic signaling, as they closely align with known mechanisms of PLOSL pathogenesis. The manuscript has been updated accordingly.

24. The presentation of docking results compares binding scores, but no comparison is made with known strong PPARG ligands beyond Rosiglitazone. Including known ligand reference values would contextualize the strength of interactions.

Response: Thank you for the suggestion. The authors have revisited the docking results and have added the control ligand Rosiglitazone results in the revised manuscript, along with the interactions as shown in Figure 8C

25. The Results section occasionally shifts into speculative statements (e.g., therapeutic implications) that belong in Discussion. Please separate objective findings from interpretation.

Response: The results section has been revised and such statements that were not required have been removed. However, some sentences that the authors felt were important for the continuous interpretation of the readers were retained.

RESPONSE TO REVIEWER COMMENTS

We thank the Referee for spending time and interest in our work and for helpful comments that will greatly improve the manuscript. We have checked all the general and specific comments provided by the Referee and have made all the necessary changes according to his instructions. Please refer to the yellow-highlighted sections or tracked changes in the revised manuscript.

Reviewer #2: The manuscript by Bokhari et al. presents a comprehensive computational study aimed at identifying key hub genes, pathways, and potential drug repurposing candidates for the rare and severe Nasu-Hakola disease (PLOSL). The authors employ a systems biology workflow, including differential gene expression analysis, protein-protein interaction (PPI) network construction, survival analysis, and sophisticated computational chemistry techniques like molecular docking and molecular dynamics (MD) simulations. The study's main claims are the identification of PPARG and AIF1 as central hub genes in PLOSL pathology and the proposal of AMG-131 and Elafibranor as promising therapeutic agents targeting PPARG. The significance of this work lies in its attempt to address a major therapeutic gap for a devastating rare disease using a cost-effective, bioinformatics-driven drug repurposing approach. The application of a 200 ns MD simulation to validate docking results adds considerable strength to the stability claims of the proposed drug-target complexes.

Well for authors must note that PLOS One encourages submissions across the methodological spectrum, but for a study making specific claims about therapeutic potential and biomarker identification, some form of experimental corroboration is increasingly expected to justify publication.

Response: Thank you for the valuable comment. As the paper scope is pure computational and we as a team mainly work on bioinformatics-related projects, the paper was submitted to PLOS ONE purely because it accepts computational work. The paper's conclusion can be validated by experimental validation; however, currently, we don’t have experimental facilities to validate the computational conclusions of this paper. The paper findings will speed up the PPARG as drug target as it may provide experimentalists to test this protein in a specific among the large dataset. The experimental

---

## [Decision Letter · Decision Letter 1]

3 Feb 2026

System Biology and Network-Based Approach to Identify the Therapeutic Signatures and Potential Inhibitors Against Polycystic Lipomembranous Osteodysplasia with Sclerosing Leukoencephalopathy

PONE-D-25-34800R1

Dear Dr. Alaa Abdulaziz Eisa.

We are pleased to inform you that your manuscript has been judged scientifically suitable for publication and will be formally accepted for publication once it meets all outstanding technical requirements.

Kind regards,

Academic Editor

PLOS One

Additional Editor Comments (optional):

Reviewers' comments:

Reviewer's Responses to Questions

**Comments to the Author**

1. If the authors have adequately addressed your comments raised in a previous round of review and you feel that this manuscript is now acceptable for publication, you may indicate that here to bypass the “Comments to the Author” section, enter your conflict of interest statement in the “Confidential to Editor” section, and submit your "Accept" recommendation.

Reviewer #1: All comments have been addressed

Reviewer #2: All comments have been addressed

2. Is the manuscript technically sound, and do the data support the conclusions?

Reviewer #1: Yes

Reviewer #2: Yes

3. Has the statistical analysis been performed appropriately and rigorously? 

Reviewer #1: Yes

Reviewer #2: Yes

4. Have the authors made all data underlying the findings in their manuscript fully available?

Reviewer #1: Yes

Reviewer #2: Yes

5. Is the manuscript presented in an intelligible fashion and written in standard English?

Reviewer #1: Yes

Reviewer #2: Yes

6. Review Comments to the Author

Reviewer #1: Authors have well addressed most of the reviewer comments; however, the discussion of PPARG remains general. The manuscript does not clearly establish a mechanistic link between PPARG signaling and PLOSL-specific pathology (e.g., microglial/macrophage dysfunction, lipid homeostasis, and chronic inflammation). This section should be strengthened with clearer disease-specific interpretation and supporting literature before the manuscript can be considered for publication.

Reviewer #2: The revised version is significantly clearer, more rigorous, and better framed as a computational systems biology contribution. The improvements to the figures, network analysis, and molecular simulations are particularly commendable.

Congratulations!

7. PLOS authors have the option to publish the peer review history of their article (what does this mean? ). If published, this will include your full peer review and any attached files.

**Do you want your identity to be public for this peer review?** For information about this choice, including consent withdrawal, please see our Privacy Policy .

Reviewer #1: **Yes:** Sourbh Suren Garg

Reviewer #2: **Yes:** Dr Saleem Iqbal

---

## [Editor Report · Acceptance letter]

PONE-D-25-34800R1

PLOS One

Dear Dr. Eisa,

I'm pleased to inform you that your manuscript has been deemed suitable for publication in PLOS One. Congratulations! Your manuscript is now being handed over to our production team.

Kind regards,

on behalf of

Dr. Md. Zubbair Malik

Academic Editor

PLOS One